# ROBUST UNCERTAINTY-AWARE LEARNING VIA BOLTZMANN-WEIGHTED NLL

## ABSTRACT

Uncertainty estimation is critical for deploying deep learning models in high-stakes applications such as autonomy and decision-making. While prior works on data uncertainty modeling estimate aleatoric uncertainty by minimizing the negative log-likelihood (NLL) loss, they often fail in the presence of outliers. To address this limitation, we introduce Robust-NLL, a drop-in replacement for vanilla NLL that filters noisy or adversarial samples. Robust-NLL learns robust uncertainty estimates in neural networks through a Boltzmann-weighted NLL loss that requires no architectural changes, additional parameters, or iterative procedures, and acts as a plug-and-play loss function that maintains full differentiability and mini-batch compatibility. We evaluate our approach on synthetic regression tasks and real-world visual localization benchmarks with injected outliers. Experimental results demonstrate that simply replacing NLL with Robust-NLL consistently improves both prediction accuracy and reliability of uncertainty estimates, achieving substantial performance gains across diverse tasks and architectures.

## 1 INTRODUCTION

Uncertainty estimation plays a crucial role in deep learning methods, especially in high-stakes applications such as autonomous navigation and medical decision-making. In these domains, it is not sufficient for a model to simply make accurate predictions; it must also quantify how confident it is in those predictions. A well-calibrated uncertainty estimate allows the system to detect ambiguous, noisy, or out-of-distribution inputs, enabling failure detection and more trustworthy decisions, which are essential for robust deployment in real-world environments.

Recent work, such as the uncertainty-aware regression framework by Kendall & Gal (2017), has made progress in modeling data uncertainty by minimizing a negative log-likelihood (NLL) loss. However, such methods remain vulnerable to outliers in training data (Detlefsen et al., 2019; Seitzer et al., 2022), which can severely distort both predictive outputs and associated uncertainty estimates. When supervised with noisy labels, these models often produce overconfident and inaccurate uncertainty estimates, limiting their practical reliability.

Despite many visual localization pipelines having adopted uncertainty-aware learning (Kendall & Cipolla, 2017; Brahmbhatt et al., 2018; Wang et al., 2020; Qiao et al., 2023; Xiao et al., 2024), existing state-of-the-art models such as `marepo` (Chen et al., 2024b) do not model uncertainty at all. In principle, adding uncertainty modeling on top of these systems should improve robustness to corrupted supervision and enhance reliability. However, we observed that using vanilla NLL loss often degrades performance in the presence of outliers (Detlefsen et al., 2019; Seitzer et al., 2022). See Figure 1 for an illustrative example.

This motivates the need for a robust uncertainty-aware learning framework that is compatible with existing architectures and improves both prediction accuracy and uncertainty quality under noisy supervision. Despite various methods being proposed for robust learning models (Barron, 2019; Shen & Sanghavi, 2019; Li et al., 2020; Menon et al., 2020; Song et al., 2023; Talak et al., 2025), robust uncertainty-aware learning remains an open research problem. Our goal is to build such a framework that can be seamlessly integrated into state-of-the-art models without any implementation changes while addressing their vulnerability to outlier corruption.

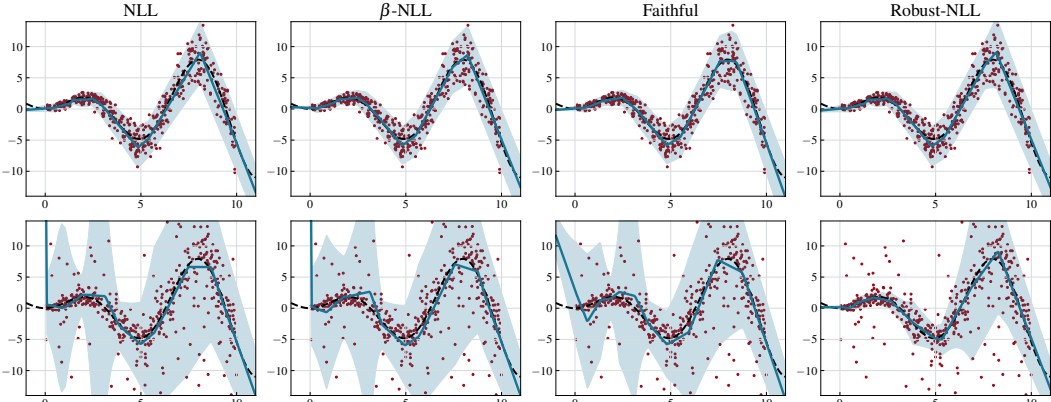

Figure 1: Comparison between uncertainty-aware methods, where the bottom row includes 25% outliers. Dotted black line denotes the ground truth; blue line and shaded area represent the predictive mean $\mu_\theta(x)$ and its 95% confidence interval $\pm 2\sigma_\theta(x)$, respectively. Our proposed Robust-NLL is the only method that remains robust to outliers, producing reliable predictions and calibrated uncertainty estimates (see Section 4.2).

To this end, we propose **Robust-NLL**, a fully differentiable Boltzmann-weighted NLL that improves robustness to outliers during training. Instead of treating all samples equally, our approach applies a smooth, differentiable sample-wise weight for each NLL loss. This ensures that Robust-NLL integrates naturally into gradient-based optimization and is compatible with mini-batch training regimes. Intuitively, samples with abnormally large losses, which often correspond to outliers, are down-weighted through a temperature-controlled Boltzmann weight, thereby reducing their influence on gradient updates. We evaluate Robust-NLL on both synthetic regression tasks and a real-world visual localization benchmark. In both cases, simply replacing vanilla NLL with our method consistently improves prediction accuracy and uncertainty calibration under label corruption, outperforming standard NLL baselines. Our main contributions are summarized as follows:

- We propose Robust-NLL, a simple yet effective robust loss formulation that is fully compatible with existing uncertainty-aware learning models.
- A theoretical connection between Robust-NLL and robust estimation is established, showing that the Boltzmann-weighting allows the method to adjust its robustness to varying outlier rates.
- Experimental results demonstrate that Robust-NLL improves both accuracy and uncertainty reliability under noisy supervision, without requiring architectural changes or additional model complexity.

## 2 RELATED WORK

### 2.1 UNCERTAINTY-AWARE LEARNING

Uncertainty is typically divided into two types: epistemic uncertainty and aleatoric uncertainty (Kiureghian & Ditlevsen, 2009). Epistemic uncertainty represents model uncertainty, which is reducible given enough training data. Common methods include MC-dropout (Gal & Ghahramani, 2016; Kendall & Gal, 2017) and ensembling (Lakshminarayanan et al., 2017), both of which require multiple forward passes to approximate the posterior distribution over model parameters. Aleatoric uncertainty, on the other hand, represents data uncertainty and is irreducible regardless of the amount of training data. The standard approach fits a Gaussian model by minimizing the negative log-likelihood (NLL) (Nix & Weigend, 1994; Kendall & Gal, 2017). However, this formulation can lead to overconfident uncertainty estimates (Guo et al., 2017; Detlefsen et al., 2019). To address this, Seitzer et al. (2022) proposed $\beta$-NLL, introducing a surrogate parameter that regularizes predictive uncertainty. Stirn et al. (2023) proposed a *faithful* constraint to ensure performance guarantees by modifying the gradient calculation. Immer et al. (2023) learns natural parameters instead of

mean and variance, which improves numerical stability during training. Despite these advances in uncertainty-aware learning, uncertainty estimation under noisy supervision remains understudied. In this work, we focus on modeling aleatoric uncertainty, with comparisons primarily against NLL and its variants.

## 2.2 ROBUST LEARNING

Robust estimation has been widely studied and applied to various tasks in computer vision (Fitzgibbon, 2003; Yang et al., 2020a) and robotics (Yang et al., 2021; Chen et al., 2024a). In general, robust estimation is often solved by iterative reweighted least squares (IRLS) (Aftab & Hartley, 2015; MacTavish & Barfoot, 2015; Ochs et al., 2015), a method that can be traced back to Weiszfeld (1937). Variants such as graduated non-convexity (Blake & Zisserman, 1987; Yang et al., 2020a; Peng et al., 2023) leverage homotopy optimization, which introduces a surrogate parameter into the IRLS scheme to address the non-convexity and initialization issue in robust estimation. For learning-based models, Shen & Sanghavi (2019) proposed a truncated robust loss with an iterative training scheme, while Barron (2019) presented a generalized loss for various robust functions. Recently, Talak et al. (2025) integrates the IRLS procedure into learning-based models and can be applied to a variety of robust functions. Other robust training strategies include gradient clipping (Menon et al., 2020; Mai & Johansson, 2021), which mitigates the influence of outlier-induced exploding gradients. Li et al. (2020) proposed a semi-supervised approach by fitting a Gaussian mixture model, as the mixture model can also be viewed as a robustified function (Olson & Agarwal, 2013) to distinguish clean and noisy samples. However, these approaches typically require iterative schemes that complicate integration into standard training pipelines. We provide a simple, differentiable robust method that can be seamlessly adopted without modifying existing optimization procedures.

## 2.3 VISUAL LOCALIZATION

There are two main modeling approaches for visual localization: relative pose regression (RPR) and absolute pose regression (APR). Early RPR methods (Wang et al., 2017; Li et al., 2018; Yang et al., 2020b) predict the relative pose (odometry) between two images but suffer from accumulated drifting errors. Recent RPR approaches (Teed & Deng, 2021; Teed et al., 2023) improve performance by learning optical flows and incorporating differentiable bundle adjustment layers. In contrast, APR methods (Kendall & Cipolla, 2017; Brahmbhatt et al., 2018) directly learn the absolute pose of a specific scene. Although efficient, APR methods lack generalization to unseen views, and multi-scene APR models (Shavit et al., 2021; Lee et al., 2024) extend this paradigm to handle multiple environments by training shared networks across environments. The performance of APR methods has further improved with the rise of attention mechanisms (Wang et al., 2020; Qiao et al., 2023). Another approach to improve APR performance has become prominent by learning scene representations (Moreau et al., 2021; Chen et al., 2022; Brachmann & Rother, 2022), but they typically require significant computational resources. Recent developments (Brachmann et al., 2023; Chen et al., 2024b) have significantly reduced training time while maintaining benchmark performance. Despite these advances, most state-of-the-art visual localization methods lack uncertainty quantification, limiting their reliability in safety-critical applications where failure detection is essential.

## 3 ROBUST UNCERTAINTY-AWARE LEARNING

Given a dataset with input $x = \{x_i \in \mathbb{R}^d\}_{i=1}^N$ and target $y = \{y_i \in \mathbb{R}\}_{i=1}^N$, we assume that the targets $y_i$ are conditionally Gaussian with the density $p(y \mid x) = \mathcal{N}(y; \mu(x), \sigma^2(x))$. The functions $\mu : \mathbb{R}^d \to \mathbb{R}$ and $\sigma^2 : \mathbb{R}^d \to \mathbb{R}$ represent the mean and variance, respectively. We estimate $\mu(x)$ and $\sigma^2(x)$ by $\mu_\theta(x)$ and $\sigma_\theta^2(x)$ with some neural network $f_\theta$ parameterized by $\theta$. The model parameters are then learned via maximum likelihood estimation by minimizing a NLL loss (Nix & Weigend, 1994; Kendall & Gal, 2017):

$$\mathcal{L}_{\text{NLL}}(\theta) = \frac{1}{N} \sum_{i=1}^N \underbrace{\frac{1}{2} \log(\sigma_\theta^2(x_i)) + \frac{\|\mu_\theta(x_i) - y_i\|^2}{2\sigma_\theta^2(x_i)}}_{\mathcal{L}_{\text{NLL}}^{(i)}(\theta)} . \quad (1)$$

This is the most common approach for uncertainty-aware regression. The first term penalizes high uncertainty, while the second encourages accuracy scaled by predictive variance. However, much like the mean square error is sensitive to outliers in normal regression, Equation 1 is vulnerable to outliers (Detlefsen et al., 2019; Seitzer et al., 2022). Therefore, we next introduce a robust weighting scheme for the NLL loss that serves as a drop-in replacement for Equation 1.

### 3.1 BOLTZMANN-WEIGHTED NLL

A natural way to improve robustness is to reduce the influence of high-loss samples during training. Inspired by max-mixtures (Olson & Agarwal, 2013), we initially considered a max operator that acts as a selector to filter high-NLL components. While this hardmax strategy completely removes the effect of large-loss outliers, it tends to be overly aggressive and rejects a large portion of the dataset. To address this, we replace hardmax with a smooth, differentiable softmax to re-weight over samples:

$$\mathcal{L}_{\text{Robust-NLL}}(\theta) = \sum_{i=1}^{N} \underbrace{w_i(\theta)\mathcal{L}_{\text{NLL}}^{(i)}(\theta)}_{\mathcal{L}_{\text{Robust-NLL}}^{(i)}(\theta)}, \tag{2}$$

$$w_i(\theta) = \frac{\exp(-\mathcal{L}_{\text{NLL}}^{(i)}(\theta)/T)}{\sum_{j=1}^{N} \exp(-\mathcal{L}_{\text{NLL}}^{(j)}(\theta)/T)}. \tag{3}$$

We further introduce a surrogate parameter $T$ that controls the inlier region. Figure 2 shows the weight function with respect to different surrogate $T$. As $T \to 0$, the weight function emphasizes the lowest NLL (with a smaller inlier region), which approximates a hardmax. On the other hand, as $T \to \infty$, all weights are nearly uniform (with a larger inlier region), reducing the loss to vanilla NLL. In particular, $w_i$ is the softmax given $T = 1$. Thus, $T$ serves as a robustness knob between uniform weighting and aggressive outlier rejection, giving us a flexible way to adapt the robustness of the method to different noise levels in the data. This idea is naturally connected to the Boltzmann distribution: the loss function $\mathcal{L}_{\text{NLL}}$ can be viewed as an energy function over data points, where the probability of an energy state is in the form $w_i(\theta) \propto \exp(-\mathcal{L}_{\text{NLL}}^{(i)}(\theta)/T)$, and $T$ is the temperature parameter that controls the sharpness of this probability. Our proposed robust loss is then defined as the expectation $\mathcal{L}_{\text{robust-NLL}}(\theta) := \mathbb{E}[\mathcal{L}_{\text{NLL}}(\theta)]$.

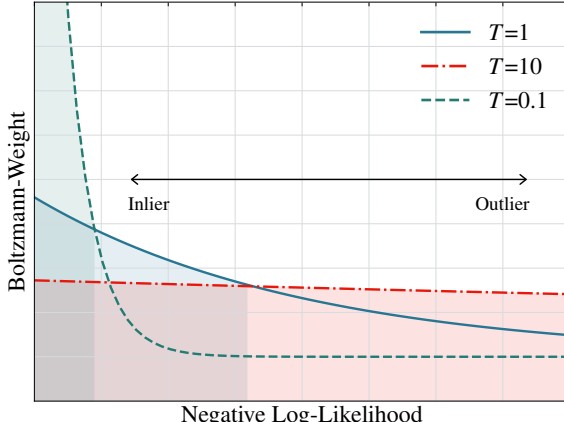

Figure 2: Example of Boltzmann distributed weight functions. The inlier (shaded) region gets larger as the surrogate parameter $T$ increases.

This formulation resembles a robust estimation of the NLL loss, where samples with higher NLL receive exponentially smaller weight. Unlike hard thresholding or clipping, this allows for smooth filtering that is fully differentiable and compatible to mini-batch settings. In contrast to many robust learning methods that rely on iterative scheme (Shen & Sanghavi, 2019; Talak et al., 2025) (e.g., an inner iteration for model fitting and an outer iteration for robust estimation), our method

integrates seamlessly into standard gradient-based training without requiring additional computation or iterative procedures. In practice, one can simply replace $\mathcal{L}_{\text{NLL}}$ by $\mathcal{L}_{\text{Robust-NLL}}$ without any other modifications, making it efficient, scalable, and easy to adopt.

## 3.2 THEORETICAL ANALYSIS

While standard NLL-based uncertainty-aware regression corresponds to the MLE solution, our method adapts the likelihood aggregation by applying a Boltzmann-weighted filter over batch NLLs. This shifts the optimization toward the inlier region in each batch, effectively reducing the influence of outliers without requiring explicit detection or hard truncation. The resulting surrogate objective remains differentiable and is compatible with gradient-based optimization.

We first write the partial derivatives of $\mathcal{L}_{\text{NLL}}^{(i)}$ with respect to mean and variance:

$$\nabla_\mu \mathcal{L}_{\text{NLL}}^{(i)}(\theta) = \frac{\mu_\theta(x_i) - y_i}{\sigma_\theta^2(x_i)}, \tag{4}$$

$$\nabla_{\sigma^2} \mathcal{L}_{\text{NLL}}^{(i)}(\theta) = \frac{\sigma_\theta^2(x_i) - \|\mu_\theta(x_i) - y_i\|^2}{2\sigma_\theta^4(x_i)}. \tag{5}$$

The derivatives of $\mathcal{L}_{\text{NLL}}$ with respect to $\theta$ is then obtained by applying standard backpropagation through the network:

$$\nabla_\theta \mathcal{L}_{\text{NLL}}^{(i)}(\theta) = \nabla_\mu \mathcal{L}_{\text{NLL}}^{(i)}(\theta)\nabla_\theta \mu_\theta(x_i) + \nabla_{\sigma^2}\mathcal{L}_{\text{NLL}}^{(i)}(\theta)\nabla_\theta \sigma_\theta^2(x_i).$$

Note that the derivatives of the weights $w_i$ with respect to $\theta$ are:

$$\nabla_\theta w_i(\theta) = \frac{1}{T}\sum_{j=1}^{N} w_i(\theta)(w_j(\theta) - \delta_{ij})\nabla_\theta \mathcal{L}_{\text{NLL}}^{(j)}(\theta),$$

we have the derivative of $\mathcal{L}_{\text{Robust-NLL}}$ with respect to $\theta$

$$\nabla_\theta \mathcal{L}_{\text{Robust-NLL}}(\theta) = \sum_{i=1}^{N}\left(w_i(\theta)\nabla_\theta \mathcal{L}_{\text{NLL}}^{(i)}(\theta) + \mathcal{L}_{\text{NLL}}^{(i)}(\theta)\nabla_\theta w_i(\theta)\right)$$

$$= \sum_{i=1}^{N} w_i(\theta)\nabla_\theta \mathcal{L}_{\text{NLL}}^{(i)}(\theta) + \frac{1}{T}\sum_{i=1}^{N} w_i(\theta)\mathcal{L}_{\text{NLL}}^{(i)}(\theta)\left(\sum_{j=1}^{N}(w_j(\theta) - \delta_{ij})\nabla_\theta \mathcal{L}_{\text{NLL}}^{(j)}(\theta)\right)$$

$$= \sum_{j=1}^{N} w_j(\theta)\nabla_\theta \mathcal{L}_{\text{NLL}}^{(j)}(\theta) + \sum_{j=1}^{N}\left(\frac{1}{T}\sum_{i=1}^{N} w_i(\theta)(w_j(\theta) - \delta_{ij})\mathcal{L}_{\text{NLL}}^{(i)}(\theta)\right)\nabla_\theta \mathcal{L}_{\text{NLL}}^{(j)}(\theta)$$

$$= \sum_{j=1}^{N}\left(w_j(\theta) + \frac{1}{T}\sum_{i=1}^{N} w_i(\theta)(w_j(\theta) - \delta_{ij})\mathcal{L}_{\text{NLL}}^{(i)}(\theta)\right)\nabla_\theta \mathcal{L}_{\text{NLL}}^{(j)}(\theta). \tag{6}$$

Since $\mathcal{L}_{\text{robust-NLL}}$ is fully differentiable, we follow the standard non-convex convergence analysis of mini-batch SGD (Bottou et al., 2018; Ghadimi & Lan, 2013).

**Lemma 1.** *Assume that $\mathcal{L}_{\text{robust-NLL}}$ is bounded below, and the stochastic gradient is an unbiased estimator of $\nabla_\theta \mathcal{L}_{\text{robust-NLL}}$ with bounded variance. If $\mathcal{L}_{\text{robust-NLL}}$ is continuously differentiable, $\nabla_\theta \mathcal{L}_{\text{robust-NLL}}$ is Lipschitz continuous, then with diminishing stepsizes (learning rates),*

$$\liminf_{t\to\infty} \mathbb{E}[\|\nabla_\theta \mathcal{L}_{\text{Robust-NLL}}(\theta)\|^2] = 0.$$

In other words, let $\{\theta_{t_k}\}_{k=1}^{\infty}$ be a subsequence of iterates $\{\theta_t\}_{t=1}^{\infty}$ produced by mini-batch SGD with the loss $\mathcal{L}_{\text{Robust-NLL}}$, the subsequence converges to a stationary point in expectation.

## 3.3 WEIGHTING GRADIENTS OF NLL

One problem of training with vanilla NLL is that the gradients are highly dependent on the predictive variance (the denominators of Equation 4 and Equation 5). Seitzer et al. (2022) introduced a

*variance-weighting* to regularize the predictive variance during training. As shown in Equation 6, our Robust-NLL introduces a similar but more general weighting factor:

$$\alpha_j := w_j(\theta) + \frac{1}{T} \sum_{i=1}^{N} w_i(\theta)(w_j(\theta) - \delta_{ij})\mathcal{L}_{\text{NLL}}^{(i)}(\theta), \tag{7}$$

such that the partial derivatives with respect to mean and variance are scaled by $\alpha_j$:

$$\nabla_\mu \mathcal{L}_{\text{Robust-NLL}}^{(j)}(\theta) = \alpha_j \cdot \frac{\mu_\theta(x_j) - y_j}{\sigma_\theta^2(x_j)}, \tag{8}$$

$$\nabla_{\sigma^2} \mathcal{L}_{\text{Robust-NLL}}^{(j)}(\theta) = \alpha_j \cdot \frac{\sigma_\theta^2(x_j) - \|\mu_\theta(x_j) - y_j\|^2}{2\sigma_\theta^4(x_j)}. \tag{9}$$

**Uniform weighting** When $T \to \infty$, the second term of Equation 7 vanishes, thus $\alpha_j \to 1/N$ since $w_j \to 1/N$ as $T \to \infty$. This reduces Equation 8 and Equation 9 to Equation 4 and Equation 5 up to a constant, respectively. In this regime, our Robust-NLL degenerates to vanilla NLL.

**Softmax weighting** When $T = 1$, the weighting factor becomes

$$\alpha_j = w_j(\theta)\Big(1 + \sum_{i=1}^{N} w_i(\theta)\mathcal{L}_{\text{NLL}}^{(i)}(\theta) - \mathcal{L}_{\text{NLL}}^{(j)}(\theta)\Big).$$

Here, the expectation value $\sum_i w_i \mathcal{L}^{(i)}$ serves as a threshold: if $\mathcal{L}^{(j)} < \sum_i w_i \mathcal{L}^{(i)}$ (likely inliers), the weighting term $\alpha_j > w_j$ is amplified; and if $\mathcal{L}^{(j)} > \sum_i w_i \mathcal{L}^{(i)}$ (likely outliers), the weighting term $\alpha_j < w_j$ is suppressed.

**Hardmax weighting** When $T \to 0$, let $j^* = \arg\min_j \mathcal{L}_{\text{NLL}}^{(j)}(\theta)$, then $w_{j^*} \to 1$ and all other $w_j \to 0$ exponentially fast. In this limit, the second term of Equation 7 also vanishes; therefore $\alpha_j \to 1$ if $j = j^*$, otherwise $\alpha_j \to 0$. This corresponds to a hardmax behavior in which only the smallest loss contributes to the gradient.

These analyses align with the intuition behind our formulation: $w_i$ acts as the weight applied to each loss component, while $\alpha_i$ can be interpreted as the weight on the gradient of each loss component. As $T$ decreases, $w_i$ concentrates on fewer samples, and $\alpha_i$ correspondingly focuses gradient contributions on fewer samples, shrinking the inlier region. As $T$ increases, both $w_i$ and $\alpha_i$ spread more evenly across the dataset, enlarging the inlier region. Choosing an optimal surrogate parameter remains an open question, since both outlier ratio and batch size $N$ are involved in determining $T$. In practice, we recommend starting with a relatively large $T$ (which reduces our Robust-NLL to vanilla NLL and guarantees baseline performance), then gradually decrease $T$ to enhance robustness.

### 3.4 ROBUSTNESS BEYOND NLL

While our method is primarily motivated by improving robustness in uncertainty-aware learning via NLL, the core idea of weighting per-sample losses using a Boltzmann distribution is not restricted to NLL-based objectives. In fact, this formulation can be applied to any differentiable loss function where individual loss values can be computed. Since the Boltzmann-weighted formulation preserves differentiability and is fully compatible with gradient-based optimization, our robust technique is a plug-and-play mechanism for robustifying a wide range of learning objectives. This approach allows us to transform any existing loss into a robust version by down-weighting high-loss (potentially outlier) samples during optimization. For example, it can be naturally combined with standard regression losses such as mean squared error (MSE), or uncertainty-aware losses like $\beta$-NLL, without requiring any change to the model architecture or training procedure. We demonstrate this flexibility in Section 4.1, validating its general applicability beyond vanilla NLL.

### 4 EXPERIMENTS

#### 4.1 LINEAR REGRESSION

We first consider a simple linear regression example $y = w^\top x + \epsilon + o$ from Talak et al. (2025), where $w \sim \mathcal{N}(0, 1)$ is the weight, $\epsilon \sim \mathcal{N}(0, 0.1)$ is a noise term, $o \sim \mathcal{N}(0, 10)$ is an outlier term, and $x$ is

uniformly sampled from $(0, 1]$. Each network directly predicts the output mean (and variance) with a single fully connected layer, trained with $10^4$ iterations and the SGD optimizer with learning rate $10^{-4}$ and batch size 16. We evaluate the mean squared error between the estimated weight $\hat{w}$ and ground truth weight $w$, with outlier rates ranging from 10% to 90%. The results are averaged over 20 Monte Carlo runs.

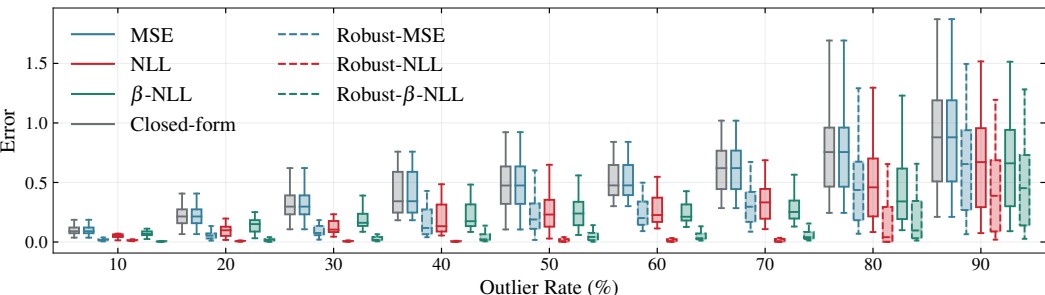

Figure 3: Evaluation of weight estimation error under increasing outlier ratios (10%–90%). We compare different losses and their robust counterparts using our proposed Boltzmann-weighted formulation. The robust versions consistently improve performance, reducing the impact of outliers.

We compare the following solutions in Figure 3: closed-form solution $\hat{w} = (X^\top X)^{-1} X^\top Y$, MSE loss, NLL loss (Nix & Weigend, 1994; Kendall & Gal, 2017), $\beta$-NLL loss (Seitzer et al., 2022), and our proposed robust scheme applied to each of these losses with the surrogate parameter $T = 1$, respectively. For fair comparison, we initialize the weight parameters of each network as the closed-form solution. This corresponds to the common *warm-up* strategy in uncertainty-aware learning: first train a mean-only network by freezing the parameters that do not affect $\mu_\theta$, then jointly optimize mean and variance (i.e., full parameter space $\theta$) together. The results show that MSE struggles with outliers and converges to some local point since the MSE loss is vulnerable against outliers. In contrast, our Boltzmann-weighting technique consistently improves performance across all losses: Robust-MSE, Robust-NLL, and Robust-$\beta$-NLL has an average relative improvement of 55.4%, 80.5%, 71.1% compared with MSE, NLL, and $\beta$-NLL, respectively. Since the vanilla NLL is widely used in the literature (as discussed in Section 2), we focus on Robust-NLL in the remainder of our experiments. Nevertheless, we reiterate that our robust training framework is loss-agnostic and can be applied to a broad class of differentiable loss functions beyond NLL.

## 4.2 SINUSOIDAL REGRESSION

We compare uncertainty-aware learning methods on the sinusoidal example from Detlefsen et al. (2019), a sine curve with an increasing amplitude of noise (Figure 1):

$$y = x \sin(x) + 0.3\epsilon_1 + 0.3x\epsilon_2, \tag{10}$$

where $\epsilon_1, \epsilon_2 \sim \mathcal{N}(0, 1)$ and $x$ are uniformly sampled from $(0, 10]$. The first additive term $\epsilon_1$ accounts for *homoscedastic* (data-independent) uncertainty, while the second term $\epsilon_2$ accounts for *heteroscedastic* (data-dependent) uncertainty since the term increases for larger $x$. All networks consist of a single hidden layer with 50 neurons with the ReLU activation function, trained with $10^5$ iterations and the Adam optimizer with a learning rate of 0.01 and batch size 256. These settings are identical to (Detlefsen et al., 2019).

The top row of Figure 1 compares the predictive mean and variance of NLL (Nix & Weigend, 1994; Kendall & Gal, 2017), $\beta$-NLL (Seitzer et al., 2022), Faithful (Stirn et al., 2023), and our Robust-NLL with the surrogate parameter $T = 64$. We use a larger $T$ in this setting because the data contains no explicit outliers; a small $T$ would make the weight function emphasize few data, degrading the performance. Note that as $T \to \infty$, Robust-NLL should have the same result as vanilla NLL. The bottom row introduces 25% of outliers by adding a noise $o \sim \mathcal{N}(0, 10)$ to Equation 10. We reduce the surrogate parameter to $T = 16$ in this case, and our Robust-NLL is the only one that remains truly robust to outliers.

## 4.3 7-SCENES VISUAL LOCALIZATION

We evaluate our proposed Robust-NLL on the 7-Scenes (Shotton et al., 2013) indoor benchmark dataset using marepo (Chen et al., 2024b) as our backbone model with an additional uncertainty head[1]. We initialize the network with the pretrained weights provided by the authors, which serve as a warm-up strategy for the mean-only network. We adopt a two-step training procedure: first train the uncertainty head using vanilla NLL loss for 100 epochs while fixing the mean-only network; then jointly finetune the entire model using our Robust-NLL loss for another 100 epochs. In addition, we apply online data augmentation by applying random rotation and scaling up to $15°$ and $1.5$ times to the images and their corresponding poses, respectively. We also apply random rotation and translation up to $15°$ and 1 meter to the predicted scene coordinates and their corresponding poses, respectively. Please refer to Chen et al. (2024b) for more details on data augmentation. The surrogate parameter is set to $T = 1$ in all scenes.

Table 1: Comparison between APR methods on 7-Scenes, reported as median error (deg/cm).

| Method | Chess | Fire | Heads | Office | Pumpkin | Kitchen | Stairs | Average |
|---|---|---|---|---|---|---|---|---|
| PoseNet | 4.48/13 | 11.3/27 | 13.0/17 | 5.55/19 | 4.75/26 | 5.35/23 | 12.4/35 | 8.12/22 |
| MapNet | 3.25/8 | 11.7/27 | 13.3/18 | 5.15/17 | 4.02/22 | 4.93/23 | 12.1/30 | 7.78/20 |
| AtLoc | 4.07/10 | 11.4/25 | 11.8/16 | 5.34/17 | 4.37/21 | 5.42/23 | 10.5/26 | 7.56/20 |
| MST | 4.66/11 | 9.6/20 | 12.2/14 | 5.66/17 | 4.44/18 | 5.94/17 | 8.45/26 | 7.28/18 |
| ActMST | 4.15/10 | 8.79/24 | 11.6/14 | 5.28/17 | 3.48/17 | 5.62/17 | 7.58/22 | 6.64/17 |
| TransAPR | 3.4/8 | 8.41/21 | 9.51/14 | 5.52/17 | 4.07/18 | 4.65/19 | 8.45/23 | 6.29/17 |
| LENS | 1.3/3 | 3.7/10 | 5.8/7 | 1.9/7 | 2.2/8 | 2.2/9 | 3.6/14 | 3.0/8 |
| DFNet | 1.48/4 | 2.16/4 | 1.82/3 | 2.01/7 | 2.26/9 | 2.42/9 | 3.31/14 | 2.21/7 |
| marepo | 1.35/2.6 | 1.42/2.5 | 2.21/2.3 | 1.44/3.6 | 1.55/4.2 | 1.99/5.1 | 1.83/6.7 | 1.68/3.9 |
| marepo$_S$ | 1.24/2.1 | 1.39/**2.3** | 2.03/1.8 | 1.26/2.8 | 1.48/3.5 | 1.71/4.2 | 1.67/5.6 | 1.54/3.2 |
| Robust-NLL | **1.11/1.9** | **1.36**/2.4 | **1.95/1.5** | **1.22/2.8** | **1.39/3.5** | **1.60/3.5** | **1.63/4.9** | **1.46/2.9** |

Table 1 summarizes our result compared to APR methods: PoseNet (2017), MapNet (2018), At-Loc (2020), and TransAPR (2023); multi-scene APR methods: MST (2021) and ActMST (2024); scene-enhanced APR methods: LENS (2021), DFNet (2022), and marepo (2024b). Note that marepo is trained without 7-Scenes, while marepo$_S$ and our Robust-NLL are finetuned with 7-Scenes. Despite using the same architecture and finetune process, our method not only provides aleatoric uncertainty estimates but also reduces the average rotation and translation error by 5.19% and 9.38%, respectively. This demonstrates the ability to improve state-of-the-art model solely by modifying the loss function, showcasing the plug-and-play nature of our approach.

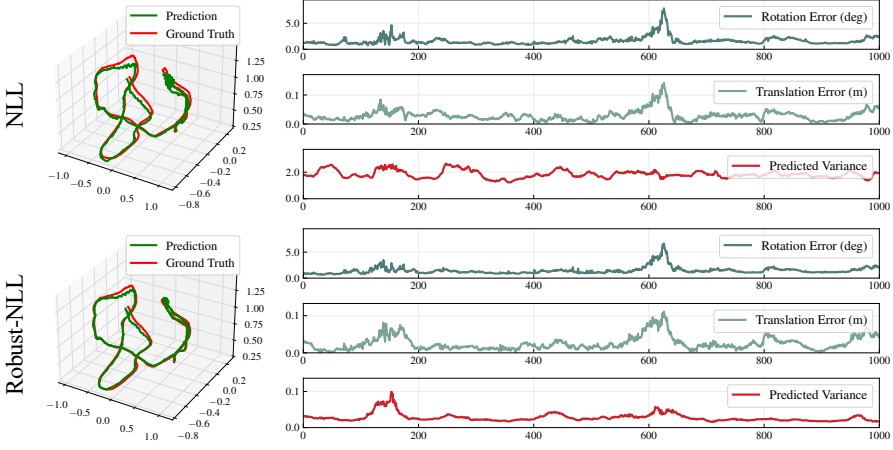

Figure 4: Example of predicted trajectory (mean), pose errors, and predicted uncertainty (variance). Robust-NLL produces uncertainty estimates that better align with actual errors.

---

[1] Code available at: https://anonymous.4open.science/r/marepo-robust.

To further validate robustness under challenging conditions, we evaluate our method in the presence of outlier labels by randomly replacing 50% of the training poses with incorrect poses. The test set remains unchanged to evaluate generalization. We compare uncertainty-aware methods with the same two-step training procedure, varying only the loss function used in the second stage. Figure 4 visualizes the predicted variance and error comparison for the scene *Fire Seq-04*. Notably, our Robust-NLL produces more reliable uncertainty estimates that better align with actual prediction errors compared to vanilla NLL. As shown in Table 2, all baselines suffer from significant performance degradation due to outlier corruption. In contrast, our Robust-NLL consistently outperforms the baselines across *all scenes*, with substantial improvements of 17.3% and 28.9% in average rotation and translation error compared to vanilla NLL. Remarkably, our Robust-NLL finetuned on the outlier-corrupted data even surpasses marepo$_S$ finetuned on clean data (Table 1), demonstrating the strong outlier tolerance of our robust formulation. These results highlight the practical utility of our robust loss formulation in real-world scenarios, where noisy supervisions are often unavoidable.

Table 2: Comparison between uncertainty-aware methods on 7-Scenes with injected 50% outliers.

| Method | Error | Chess | Fire | Heads | Office | Pumpkin | Kitchen | Stairs | Average |
|---|---|---|---|---|---|---|---|---|---|
| NLL | Rotation (deg) | 1.261 | 1.435 | 2.293 | 1.401 | 2.661 | 1.753 | 1.738 | 1.792 |
| | Translation (m) | 0.026 | 0.025 | 0.025 | 0.032 | 0.104 | 0.044 | 0.057 | 0.045 |
| $\beta$-NLL (2022) | Rotation (deg) | 1.266 | 1.441 | 2.228 | 1.393 | 1.747 | 1.756 | 1.739 | 1.652 |
| | Translation (m) | 0.026 | 0.025 | 0.024 | 0.031 | 0.121 | 0.043 | 0.056 | 0.047 |
| Faithful (2023) | Rotation (deg) | 1.242 | 1.439 | 2.219 | 1.333 | 2.273 | 1.697 | 1.754 | 1.708 |
| | Translation (m) | 0.022 | 0.026 | 0.022 | 0.032 | 0.082 | 0.043 | 0.055 | 0.040 |
| Robust-NLL | Rotation (deg) | **1.154** | **1.312** | **1.998** | **1.242** | **1.426** | **1.651** | **1.590** | **1.482** |
| | Translation (m) | **0.020** | **0.024** | **0.016** | **0.029** | **0.040** | **0.038** | **0.054** | **0.032** |

### 4.4 LIMITATION AND FUTURE WORK

While Robust-NLL improves both prediction accuracy and uncertainty calibration under noisy supervision, it still requires careful tuning of the surrogate parameter $T$. Selecting an appropriate $T$ can be dataset-specific and requires extensive validation, especially in real-world settings where outlier rates are often unknown. In addition, our current formulation operates at the mini-batch level, which also makes $T$ sensitive to batch size. This may limit the method's stability under small-batch or streaming scenarios. We plan to investigate adaptive strategies for learning $T$ during training such as utilizing homotopy optimization, gradually adjusting $T$ over the course of training. Finally, while we demonstrate our approach on regression tasks, future work could explore extending Robust-NLL to classification settings. Evaluation on larger and more diverse real-world datasets would further validate the method's robustness and generalization.

### 5 CONCLUSION

We presented Robust-NLL, a simple yet effective loss formulation for robust uncertainty-aware learning. By applying a Boltzmann-distributed weight to per-sample NLL values within each minibatch, our method dynamically downweights outliers during training without requiring architectural changes or additional supervision. Robust-NLL is fully differentiable and integrates naturally into standard gradient-based optimization pipelines. Through experiments on both synthetic regression and visual localization benchmarks, we demonstrated that Robust-NLL consistently improves both prediction accuracy and uncertainty calibration under noisy supervision. We believe this framework offers a practical and flexible tool for improving the reliability of uncertainty-aware models in real-world settings.

---

[1]The authors used large language model (Claude) to help polish the writing. All content was reviewed and approved by the authors.

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
