# OpenReview forum: "Robust Uncertainty-Aware Learning via Boltzmann-weighted NLL"
_ICLR.cc/2026/Conference — Submitted to ICLR 2026_

### Official Review · Reviewer_qR4J · 2025-10-31

**Soundness:** 3
**Presentation:** 3
**Contribution:** 2
**Rating:** 2
**Confidence:** 3

**Summary:**

This work presents Robust-NLL, which serves as a plug-and-play loss replacing vanilla NLL loss for robust uncertainty-aware training against label-space outliers. The proposed loss function uses softmax reweighting over sample losses to filter out outliers. The author also provides theoretical analysis and empirical verification of their proposed method.

**Strengths:**

The proposed method is clear and easy to understand. Empirical results also verify the effectiveness of the proposed method, boosting baseline results by adding the reweighting term.

**Weaknesses:**

My main concern with this work is the novelty of the proposed method. As depicted in Section 3.1, the proposed Robust-NLL loss is a regular NLL loss equipped with softmax-with-temperature weighting. With a Gaussian posterior, such a weighting function is simply determined by the Euclidean distance of the outlier label and predicted mean. The method might already be well examined by practitioners and prior works.

Nevertheless, it might be worth a detailed discussion of this type of method. However, the provided theoretical analysis covers only a limited aspect of the proposed method. It focuses on the differentiability of the Robust-NLL loss and its behavior when temperature $T=0,1,\infty$, which in general is a result one can easily expect given the formulation of the proposed loss and properties of softmax-with-temperature. A more detailed analysis, for example, deriving bounds that demonstrate the robustness against outliers of the Robust-NLL loss, would be greatly appreciated.

**Questions:**

As the authors stated in Section 4.4, selecting the optimal $T$ is critical in practice. Are there any intuitions or evidence for choosing a good $T$ given a dataset?

---

### Official Review · Reviewer_mpW2 · 2025-10-31

**Soundness:** 2
**Presentation:** 3
**Contribution:** 2
**Rating:** 4
**Confidence:** 4

**Summary:**

The authors study uncertainty estimation for regression.

They propose Robust-NLL, a simple and intuitive modification of the standard NLL loss that weighs each loss term with a softmax weight computed across the batch. Robust-NLL is supposed to make the model training more robust to outliers in the train labels.

They evaluate Robust-NLL on two synthetic 1D regression examples, and on a visual localization dataset. They compare the performance with standard NLL and two NLL variants.

**Strengths:**

- The paper is very well written and solid overall, basically not a single typo or similar issue.
- The studied problem is interesting, I still think uncertainty estimation for regression is understudied compared to classification.
- The proposed Robust-NLL method is conceptually simple and intuitive, it does indeed seem like an easy-to-use plug-in replacement for the standard NLL loss.

**Weaknesses:**

- The experimental evaluation is not particularly extensive, just a single real-world dataset.
- As stated by the authors themselves, Robust-NLL "_requires careful tuning of the surrogate parameter T. Selecting an appropriate T can be dataset-specific and requires extensive validation, especially in real-world settings where outlier rates are often unknown_".
- I think the experimental results might be somewhat misleading, given that the results for Robust-NLL in Figure 1 are obtained with two different values of T (T=64 in the top row, T=16 in the bottom row), and that Robust-NLL not is compared with standard NLL on the original train dataset in Section 4.3.
- Section 3.1 and 3.2 contain quite a lot of details which I think could be better suited for an appendix. This would create space for an extended experimental evaluation.
- The technical contribution/novelty is perhaps somewhat limited, but on the other hand, the simplicity of Robust-NLL is also a strength.

**Questions:**

Questions/suggestions:
- What would the results for Robust-NLL look like in Figure 1 if the same T was used in both rows? Is it not quite unfair to compare standard NLL with Robust-NLL using two different values of T here?
- Why do you not compare Robust-NLL with NLL, $\beta$-NLL and Faithful also in Table 1, with the original train data? Does it actually outperform standard NLL in this setting?
- I would really like to see at least one more real-world image-based regression task being added to the experimental evaluation.
- You mention that the Robust-NLL approach in principle could be applied also to classification, could you expand on this? If experimental results could be added also for some image classification task, that would probably strengthen the paper quite significantly.
- Section 2 is nice, but could also be made a bit shorter to make more space for the experimental evaluation.
- Could the proposed approach be extended to multi-dim (more than just 1D) regression problems as well?




Minor things:
- Line 214: "In contrast to many robust learning methods that rely on iterative scheme" --> "In contrast to many robust learning methods that rely on an iterative scheme" / "In contrast to many robust learning methods that rely on iterative schemes"?

---

### Official Review · Reviewer_CHX1 · 2025-11-03

**Soundness:** 3
**Presentation:** 3
**Contribution:** 3
**Rating:** 4
**Confidence:** 4

**Summary:**

This paper proposes a robust uncertainty-aware learning where they weight the NLL loss of each training through a temperature-dependent softmax distribution. They provide theortetical analysis of their proposed approach and demonstrate their proposed method's effiicany in three different tasks ranging from simple linear regression to visual localization.

**Strengths:**

**1. Well-written.** This paper is well-written with clearn presentation of their proposed methodology. They initiate their discussion from uncertainty-theoretic point of view and sheds light on the lack of robustness of the classical approaches. Then, they discuss their proposed method in apt details.

**2. Mix of synthetic and real dataset.** Their proposed approach outperforms many of the former method in the synthetic setting. They also demonstrate applicability of their method in visual localization as a practical regression task.

**Weaknesses:**

**1. Literature review missing** This paper proposes a weighting scheme of training samples in the training. However, this particular approach has been discussed in several previous works [1-2]. The paper needs to revisit those papers and shed lights on their proposed approach in light of those works.  [1] focuses on attention mechanism for Multiple-Instance Learning and [2] focuses on addressing noisy labels through softmax activations.

**2. More experiments needed** This paper is a direct improvement over the original uncertainty-awre regression network in [3]. Therefore, it would be best to recreate those experiments of [3] involving depth estimation and directly compare with [3]. This baseline study will clearly prove the superiority of the proposed methods' efficacy.

[1] Ilse, M., Tomczak, J. and Welling, M., 2018, July. Attention-based deep multiple instance learning. In International conference on machine learning (pp. 2127-2136). PMLR.

[2] Zhou, T., Wang, S. and Bilmes, J., 2020. Robust curriculum learning: from clean label detection to noisy label self-correction. In International conference on learning representations.

[3] A. Kendall and Y. Gal. What uncertainties do we need in bayesian deep learning for computer vision? In Advances in Neural Information Processing Systems, pp. 5574–5584, 2017.

**Questions:**

Is the proposed robustness approach applicable to image classification as well? Similar to the tasks studied in the paper, image classification also relies on NLL-based training and involves a mixture of easy and hard samples. If the proposed method can be generalized to classification settings, it could have broad applicability across domains such as image classification, token classification in Transformers, and image segmentation.

---

### Official Review · Reviewer_c95g · 2025-11-03

**Soundness:** 2
**Presentation:** 3
**Contribution:** 2
**Rating:** 4
**Confidence:** 3

**Summary:**

This paper introduces Robust-NLL, a modified loss function that improves uncertainty estimation in neural networks when training data contains outliers. The method uses Boltzmann weighting to down-weight noisy samples while maintaining compatibility with standard training procedures—requiring no architectural changes or additional parameters. Experiments on synthetic and real-world tasks show improvements in both prediction accuracy and uncertainty calibration compared to standard negative log-likelihood training.

**Strengths:**

- The paper is well written and easy to follow.
- The authors of the paper tackle an important problem.
- The proposed approach is technically sound.
- The experiments conducted show significant improvements in performances across several different scenarios.

**Weaknesses:**

- To the best of my understanding, the authors of the paper is missing some simple yet relevant baselines. For instance, what if we replaced the regular L2 loss with L1 loss or Huber loss, which are known to be more robust than L2 loss? Moreover, a comparison using the loss proposed in "A General and Adaptive Robust Loss Function" seems also relevant. How does the proposed method perform against these losses?
- The paper seems like a somewhat obvious extension of the focal loss [1] for classifications to the case of L2 loss, which also proposed to weight loss based on the probability. Similar idea of weighting loss based on the current probability values have also been extensively explored in prior literatures [2-4] in the context of noise-robust classification.
- The experiments conducted are somewhat limited. There are a lot other real world regression-based problems that can be considered to demonstrate the effectiveness.

[1] "Focal Loss for Dense Object Detection"
[2] "Normalized Loss Functions for Deep Learning with Noisy Labels"
[3] "Generalized cross entropy loss for training deep neural networks with noisy labels"
[4] "Symmetric Cross Entropy for Robust Learning With Noisy Labels"

**Questions:**

- Standard benchmark dataset seems to have been used for experiments in 4.3. Does the dataset inherently contain noise? If so, what is the noise ratio?

---

### Meta-Review · Area_Chair_bRkk · 2025-12-24

**Summary:**

This paper introduces Robust-NLL, a loss function designed to improve uncertainty estimation in neural networks by mitigating the impact of outliers. The method employs a Boltzmann-weighted negative log-likelihood loss function, which allows for robust learning without requiring architectural changes or additional parameters. The method is evaluated on synthetic regression tasks, and real-world visual localization benchmarks with injected outliers, which demonstrate improvements in both prediction accuracy and the reliability of uncertainty estimates.

Reviewers appreciated the clarity of the presentation and the importance of the problem being tackled.
Methodologically, the paper was praised for its technical soundness.

Reviewers however raised concerns in terms of a lack of contextualization within the broader literature and limited baseline comparisons to other robust uncertainty estimation methods. This also had repercussions on the perceived significance of the work, as the lack of discussions that would highlight the novelty of the proposed approach compared to existing methods which make use of similar components contributed to doubts regarding about the originality of the contribution.

Reviewers also raised concerns due to the limited empirical validation, deemed as insufficiently comprehensive to convincingly validate the proposed method, especially given the rich availability of benchmarks for uncertainty estimation in real-world regression scenarios.

The lack of rebuttal from the authors left these concerns unaddressed, ultimately leading to a consensus that while the method is technically sound and addresses an important problem, the paper falls short in terms of novelty, empirical validation, and contextualization within the existing literature.

**Reviewer Concerns:**

Rebuttals were not provided, none of the comments and concerns were addressed

**Reviewer Scores:**

| Reviewer | initial score | predicted final score |
|---:|---:|---:|
| qR4J | 2 | 2 |
| c95g | 4 | 4 |
| CHX1 | 4 | 4  |
| mpW2 | 4 | 4  |

(no change, as rebuttals were not provided)

---

### Decision · Program_Chairs · 2026-01-26

Reject